# Conventional and Molecular Techniques from Simple Breeding to Speed Breeding in Crop Plants: Recent Advances and Future Outlook

**DOI:** 10.3390/ijms21072590

**Published:** 2020-04-08

**Authors:** Sunny Ahmar, Rafaqat Ali Gill, Ki-Hong Jung, Aroosha Faheem, Muhammad Uzair Qasim, Mustansar Mubeen, Weijun Zhou

**Affiliations:** 1National Key Laboratory of Crop Genetic Improvement, College of Plant Science and Technology, Huazhong Agricultural University, Wuhan 430070, Hubei, China; sunny.ahmar@yahoo.com (S.A.); uzairqasim1149@yahoo.com (M.U.Q.); 2Oil Crops Research Institute, Chinese Academy of Agriculture Sciences, Wuhan 430070, China; drragill@caas.cn; 3Graduate School of Biotechnology & Crop Biotech Institute, Kyung Hee University, Yongin 17104, Korea; 4State Key Laboratory of Agricultural Microbiology and State Key Laboratory of Microbial Biosensor, College of Life Sciences Huazhong Agriculture University, Wuhan 430070, China; 5State Key Laboratory of Agricultural Microbiology and Provincial Key Laboratory of Plant Pathology of Hubei Province, College of Plant Science and Technology, Huazhong Agricultural University, Wuhan 430070, China; 6Institute of Crop Science and Zhejiang Key Laboratory of Crop Germplasm, Zhejiang University, Hangzhou 310058, China

**Keywords:** food security, food scarcity, conventional breeding, CRISPR/Cas9, CRISPR/Cpf1, high-throughput phenotyping, speed breeding

## Abstract

In most crop breeding programs, the rate of yield increment is insufficient to cope with the increased food demand caused by a rapidly expanding global population. In plant breeding, the development of improved crop varieties is limited by the very long crop duration. Given the many phases of crossing, selection, and testing involved in the production of new plant varieties, it can take one or two decades to create a new cultivar. One possible way of alleviating food scarcity problems and increasing food security is to develop improved plant varieties rapidly. Traditional farming methods practiced since quite some time have decreased the genetic variability of crops. To improve agronomic traits associated with yield, quality, and resistance to biotic and abiotic stresses in crop plants, several conventional and molecular approaches have been used, including genetic selection, mutagenic breeding, somaclonal variations, whole-genome sequence-based approaches, physical maps, and functional genomic tools. However, recent advances in genome editing technology using programmable nucleases, clustered regularly interspaced short palindromic repeats (CRISPR), and CRISPR-associated (Cas) proteins have opened the door to a new plant breeding era. Therefore, to increase the efficiency of crop breeding, plant breeders and researchers around the world are using novel strategies such as speed breeding, genome editing tools, and high-throughput phenotyping. In this review, we summarize recent findings on several aspects of crop breeding to describe the evolution of plant breeding practices, from traditional to modern speed breeding combined with genome editing tools, which aim to produce crop generations with desired traits annually.

## 1. Introduction

Since the early 1900s, plant breeding has played a fundamental role in ensuring food security and safety and has had a profound impact on food production all over the world [1,2]. In recent years, however, problems related to food quality and quantity globally have arisen as a consequence of the excessive food requirement for the rapidly increasing human population. Furthermore, radical changes in weather conditions caused by global climate change are causing heat and drought stress; consequently, farmers around the world are facing significant yield losses [3]. Global epidemics, such as the Irish potato blight of the 1840s and the Southern corn leaf blight in the United States in the 1970s, were disastrous events leading to the deaths of millions of people due to food shortage [4,5]. In recent years, the ratio of food production to consumption has decreased considerably, while both urbanization rates and demographic growth have increased globally. In this era of fast development and rapid growth, people prefer to consume processed foods, where nutritional quality is compromised. The world is expected to reach 10 billion by 2050, but no satisfactory strategies are in place to feed this massive population [6,7]. Developed countries have increased their agricultural productivity, partially meeting their food requirements, but this has resulted in increased stress on food manufacturing departments [8].

Plant breeding can be used to develop plants with desired traits [9]. Artificial plant selection has been used by humans for the past 10,000 years, selecting and breeding plants with higher nutritional values [10] (Figure 1). Traditional agricultural methods aimed to improve the nutritional status of different food plants. Recent scientific developments provide a wide range of possibilities and innovations in plant breeding [11]. To satisfy the continuously increasing demand for plant-based products, the current level of annual yield enhancement in major crop species (varying from 0.8–1.2%) must be doubled [12].

The introduction of Mendelian laws revolutionized the field of crop breeding. Over the last 150 years, crop development has been altered to a great extent as a consequence of contemporary cutting-edge genomics [13]. Different approaches have been used to shorten the duration of plant reproductive cycles. Novel techniques developed in this decade, such as genomic selection, high-throughput phenotyping (HTP), and modern speed breeding, have been shown to accelerate plant breeding. Genetic engineering and molecular methods have also played a role in developing crops with desirable characteristics using gene transformation [14,15,16,17]. Other techniques like large-scale sequencing, genomics, rapid gene isolation, and high-throughput molecular markers have also been proposed to improve the breeding of commercially important crop species, such as cisgenesis, intragenesis, polyploidy breeding, and mutation breeding [18,19,20,21].

Conventional breeding techniques are inadequate for plant genome enhancement to develop new plant varieties. To overcome this obstacle in plant breeding practices, molecular markers have been used since the 1990s for the selection of superior hybrid lines [22]. Improving plant phenotype for a specific desirable trait involves the artificial selection and breeding of this given trait by the plant breeder. Generally, breeders tend to focus on traits of diploid or diploid-like crops (e.g., maize and tomatoes) rather than polyploid crops (e.g., alfalfa and potatoes), which have more complex genetics. Breeders hence prefer to use crops with shorter reproductive cycles, which allow the production of several generations in a single year and, leading to faster production of the desired phenotypes by artificial breeding compared to crops that only reproduce annually or perennial plants that only reproduce every few years [23,24,25]. Plant breeding, combined with genome studies, enhances the accuracy of breeding practices and saves time [26]. Compared to other kingdoms, plants are more easily genetically manipulated to obtain desired genetic combinations by selfing, crossbreeding (or both) given their short generation time, and large population size available for analyses [27]. In the early 1980s, NASA partnered with Utah State University to explore the possibility of growing rapid cycling wheat under constant light in space stations. This joint effort resulted in the development of “USU-Apogee”, a dwarf wheat line bred for rapid cycling [28,29]. Recently, Lee Hickey and colleagues solved this issue by presenting the idea of “speed breeding”, a non-GMO path enabling researcher to turn over many generations and select plants for desired traits between many variations [8,16]. This method uses regulated environmental conditions and prolonged photoperiods to achieve between four and six generations per year of long duration crops (i.e., wheat, barley, and canola) [16,30,31].

Researchers outlined the evolving EU regulatory framework for GMOs and discussed potential ways of regulating plant varieties developed using precision breeding approaches such as clustered regularly interspaced short palindromic repeats (CRISPR), and CRISPR-associated (Cas) proteins CRISPR/Cas9 [32]. Research interest in genetically engineered crops (and more precisely “biotech crops”) has been increasing, given the urgent need to ensure food security for the growing human population [33].

Genome editing involves inserting, deleting, or substituting a foreign gene in the organism’s DNA. Upon successful transformation, this new sequence is integrated into the host genome [34,35]. Several processes are involved in the fixation of specific DNA sequences, cut with the help of nucleases. Plant breeding alone cannot achieve the required traits, but using the CRISPR-associated (Cas) enzymes (CRISPR/Cas and CRISPR/Cpf1) can help meet the needs for efficient crop research [36,37]. In this review, we discuss the use of conventional and non-conventional plant breeding techniques for different crops, as well as the use of genome editing techniques to change and improve desired phenotypes. Moreover, the potential correlations between these approaches used to develop future strategies for crop improvement will also be explored.

## 2. Mutation through Traditional or Conventional Breeding

The advantage of conventional plant breeding consists of increasing the availability of genetic resources for crop improvement through introgression of the desired traits. However, some plants are at risk of becoming susceptible to environmental stress and losing genetic diversity [38]. Thus, traditional cultivation methods are not sufficient to resolve global food security issues. Combining multiple phenotypic characters within a single plant variety would successfully increase yield and has been widely used, however, new breeding techniques are less expensive and will enable faster production of genetically improved crops [39].

In recent times, improvements in traditional plant breeding have been introduced, such as wide crosses, introgression of traits from wild relatives by hybrid breeding, mutagenesis, double haploid technology, and some tissue culture-based approaches such as embryo and ovule rescue (to achieve maximum plant regeneration) and protoplast fusion [40,41,42]. Food and feed crops developed by conventional plant breeding have specific natural phenotypic and agronomic properties. To improve crop quality, researchers have introgressed many beneficial traits through plant breeding with wild relatives, such as higher yield, abiotic and biotic stress resistance, and increased nutritional value [39,43,44]. The identification and combination of traits in familiar genotypes and the selection of high-performing varieties can establish a crop lineage with the desired properties. That being said, this approach can have potentially adverse impacts on food and environmental safety as it occasionally gives rise to safety concerns through unpredictable effects [9,45].

A trait (e.g., stress tolerance) can be improved by selecting the best hybrid progeny with the desired trait using cross breeding [46] (Figure 2a). Desired traits can also be introduced into a chosen ‘best’ recipient line through backcrossing of the selected progeny with the recipient line for several generations to reduce unwanted phenotype combinations [47]. Genetic variability can be reduced by the use of long-term traditional breeding methods; thus, the introduction of new genes is required for the improvement of desired traits by speed breeding, mutation breeding, and rapid generation advance (RGA) [16,31,48]. From this point of view, mutations could be useful in plant breeding programs and all these precision breeding tools can contribute to the improvement of specific features during the breeding cycle. Plant breeding is always approached holistically by analyzing all applicable agricultural functionality (Figure 3).

Identifying plants with desirable traits among existing plant varieties (or developing new phenotypes if these are not found naturally) is the initial and most important step in plant breeding. It would be impossible to develop new varieties or improve existing ones without natural genetic variation determined by spontaneous mutations. Ossowski et al. [49] concluded that the de novo spontaneous mutation rate was 7 × 10^−9^ base replacements per site per generation in all the nuclear genomes of five *Arabidopsis thaliana* accumulation lines sustained by single seed descent (SSD) over 30 generations. [50]. This is expected to be true for the genomes of most other plant species: for example, about 20 billion mutations occur each year in a one-hectare wheat field (personal communication with Professor Detlef Weigel, Max Planck Institute for Developmental Biology, Germany).

Another technique to improve plant varieties by conventional breeding is through mutation breeding. Mutagenesis is the phenomenon in which sudden heritable changes occur in the genetic material of an organism. It can occur spontaneously in nature or can be a result of exposure to different chemical, physical, or biological agents [51]. Mutation breeding is classified based on the three known types of mutagenesis. The first is radiation-induced mutagenesis in which mutations occur as a result of exposure to radiation (gamma rays, X-rays, or ion beams.); second is chemically induced mutagenesis; while the third is insertional mutagenesis, a consequence of DNA insertions either through the genetic transformation and insertion of T-DNA or the activation of transposable elements (i.e., site-directed mutagenesis; Table 1) [50,52]. According to Van Harten (Professor Agricultural University, Wageningen, The Netherlands), the history of plant mutation spans back to 300 BC, while the term mutation was first used in 1901 by Hugo De Vries, who reported during the final year of his studies that heredity might be changed by another mechanism, different from recombination and segregation [53]. He examined genomic variations and described them as heritable changes arising from this unique mechanism [54]. Numerous steps are required in any mutation breeding strategy: first, reducing the number of potential variants among the mutagenized seeds or other propagules for close evaluation of the first (M1) plant generation [51,54]. The benefits of mutation breeding over other breeding methods rely on the ability to select useful variant mutants in the second (M2) or third (M3) generations (Figure 2b).

Artificial mutation-causing agents are called mutagens; they are generally classified into two categories: physical and chemical mutagens [55]. They can induce mutations in almost any planting materials, including in vitro cultured cells, seedlings, and whole plants. Seeds are the most frequently used plant material for this specific purpose, but recently, various forms of plant propagules, such as tubers, bulbs, rhizomes, and mutation-induced vegetative propagated plants, are being used more frequently, as scientists take advantage of totipotency in single cells [56]. For example, with the use of ethyl methanesulfonate (EMS) and fast neutrons, collections of M82 tomato mutants were produced and more than 3000 phenotype alterations were classified [57]. An EMS-induced mutation library for the miniature dwarf tomato cultivar Micro-Tom has also been created, creating another resource for tomato genetic studies [58].

Despite considerable success during the last century, the advances in yields of major crops (e.g., wheat) stabilized or even declined in many regions of the world [61,62]. Restrictions on phenotyping efficiency are increasingly being perceived as key constraints to genetic enhancements in breeding practices [63,64]. Specifically, HTP may cause a bottleneck in traditional breeding, marker-assisted selection (MAS), or genomic selection, where phenotyping is important to establish the accuracy of statistical models [63,65]. Accurate phenotyping is also required to replicate the outcomes of mutagenesis (i.e., GMOs) [66]. Deery et al. [67] and White and Conley [68] reviewed in great detail the benefits and challenges of potential phenotyping platforms, such as HTP.

Furthermore, SSD can be accelerated through the use of HTP [69,70]. SSD is most suitable for handling large segregating populations; while HTP tools are used in breeding programs [71]. Without undermining genetic variability and genetic development, SSD optimizes resource distribution, reducing the time spent growing crops and lowering costs associated with earlier generations’ progress [72]. SSD has been successfully used in the groundnut breeding program, with the implementation of an inbreeding cycle producing multiple generations annually to advance fixed lines to multisite evaluation tests [73]. This speed breeding approach is ideal for SSD programs, particularly in cereal crops, allowing for the rapid cycling of multiple lines with healthy plants and viable seeds [30].

## 3. Mutagens for Molecular Breeding

One of the principal goals in the field of molecular biology is to identify and manipulate genes involved in human, animal, and plant disorders. Genomic tools used in such studies include restriction enzymes, biomarkers, molecular glue (ligases), as well as transcription and post-translational modification machinery [74]. Furthermore, molecular biological approaches are widely used to develop biofortified crops and plant varieties with high yield, new traits, and resistance to insect pests and diseases [75,76]. Globally, about 40 million hectares have been assigned to transgenic cultivars, which were commercialized after testing their biosafety level in 1999 [75]. Plant breeding was then reformed when researchers started to combine traditional practices with molecular tools to address phenotypic changes concerning the genotype of plant traits [77]. Accurate genome sequencing is essential before molecular tools can be used, and next-generation sequencing (NGS) allows researchers to decipher entire genomes and produce vast gene libraries for bioinformatics studies [78]. NGS opens new possibilities in phylogenetic and evolutionary studies, enabling the discovery of novel regulatory sequences and molecular markers [79]. Molecular biology is also facilitating the identification of diverse cytoplasmic male sterility sources in hybrid breeding. Some fertility restorer genes have been cloned in maize, rice, and sorghum [80]. Mutations in the target gene can be screened using target-induced local lesions in the genome (TILLING) and Eco-TILLING, which can directly identify allelic variations in the genome [81]. The most recent studies have determined the structure of plant germplasm using bulked segregant analysis [82], association mapping, genome resequencing [83,84], and fine gene mapping. This allows for the identification of single base-pair polymorphisms based on single sequence repeats, single nucleotide polymorphisms (SNPs), and unique biomarkers linked to quantitative trait loci (QTL) for genome manipulation, germplasm enhancement, and creating high-density gene libraries [85]. Traditional mutagenesis has certain limitations, as it can produce undesirable knockout mutations. It is also time-consuming and requires large-scale screening [86]. However, MAS is a direct approach for tracking mutations that improve backcrossing efficiency (or “breeding by design”) [87] and determining the homogeneity of the progeny phenotypes.

In principle, all genome cleavage techniques produce double-stranded breaks (DSBs), blunt ends, or overhangs of the target nucleotide fragment, whether by homologous recombination, site-directed insertion/substitution of genes, or knockout mutations [88]. These DSBs, produced as a result of the action of sequence-specific nucleases (SSNs), are repaired by the non-homologous end joining (NHEJ) mechanism, which adds or removes nucleotides by the homology-directed repair pathway, directing DNA substitutions at target sites [89]. Various literature reviews report three primary SSN systems for genome editing. The first involves zinc finger nucleases (ZFNs), which form the basis for DNA manipulation. The second system involves transcription activator-like effector nucleases (TALENs), while the third system, the most important revolution in cutting-edge genomics, is a clustered regularly interspaced short palindromic repeats/associated protein 9 (CRISPR/Cas9) system [88,90,91,92]. The use of ZFNs has certain limitations: the constructs are not easy to design and transform, even in plants, and it is an expensive approach. Moreover, some researchers have reported non-specific nucleotide recognition because of their origin from eukaryotic transcription motifs, making this approach less reliable for genome editing [93,94]. Most restriction nucleases are derived from bacteria and TALENs were isolated from the prokaryotic plant pathogen *Xanthomonas* [95]. However, TALENs comprise large and repetitive constructs that require a lot of time and precision to edit the target sequence [96]. Soon after the discovery of TALENs, another promising nuclease (CRISPR/Cas9) was found in a bacterial immune system [97]. This system has been widely used in recent plant genome editing studies and has started replacing the TALEN and ZFN systems due to its high efficiency and accuracy in inducing site-directed breaks in double-stranded DNA [98]. Recently, a CRISPR-associated endonuclease from *Prevotella* and *Francisella* (Cpf1) has emerged as a replacement tool for precise genome editing, including DNA-free dissection of plant material, with higher potency, specificity, and enormous possibilities of wider application [99,100]. The base-editing approach using CRISPR/nCas9 (Cas9 nickase) or dCas9 (deactivated Cas9) fused with cytidine deaminase is a powerful tool to create point mutations. In this study, we point out the remarkable *G. hirsutum*-base editor 3 (GhBE3) base enhancing system developed to create single base mutations in the allotetraploid genome of cotton (*Gossypium hirsutum*) [101].

## 4. CRISPR/Cas9 and CRISPR/Cpf1 as Genetic Dissection Tools

The CRISPR/Cas9 system is a high-throughput discovery system in cutting-edge genomics, with recent studies reporting extensive use of Cas9 in gene transformation, drug delivery, and knockout mutations based on NHEJ-mediated DSBs [102]. Several studies investigated the mode of action of this potent nuclease and discovered the presence of a CRISPR loci, a cluster of repeating nucleotides in bacterial and archaeal immune systems [103]. These loci have a unique sequence, comprising of Cas9-encoding operons, transcription machinery, and consecutive repeats originating from various viral genomes separated by spacer sequences. These repeats were incorporated into the bacterial genome either by a virus or another foreign invader following an immune reaction [17].

Yin et al. (2017) found that Cas9 can be ‘tricked’ by supplementing any foreign nucleotide sequence that is digested and inserted in the bacterial genome. To knock a gene out, the CRISPR/Cas9 system is designed accordingly and transformed to explants via *Agrobacterium*, electroporation or the biolistic method. The regenerated plantlets’ grown from the transformed callus are then transferred to planting soil [104]. The CRISPR/Cas9 gene knockout system has four significant features: (a) synthetic guide RNA (about 18–20 nucleotides) binding to target DNA, (b) Cas9 cleavage at 3–4 nucleotides after the adjacent proto spacer motif (PAM) (generally, 50 NGG identifies the PAM sequence) [105], (c) selection of a suitable binary vector and sgRNA cloning, and (d) transforming the construct in explants via *Agrobacterium* or microprojectile gene bombardment (Figure 4). *Agrobacterium*-mediated transformation is preferred in most studies given its efficiency and secure delivery [106]. The transformants are raised in growth chambers and examined for mutation studies using PCR, western blotting, ELISA, genotyping, sequencing, and other molecular techniques.

With recent advances in molecular biology and the discovery of sequences in the microbial immune system, biotechnologists can manipulate the organism’s genome in a specific and precise way with the aid of CRISPR and its associated Cas proteins. This remarkable genome editing technique is categorized into two broad classes and six types: class 1 with types I, III, and IV and class 2 with types II, type V, and type VI [107]. Type II is the most widely used system in genome editing, while CRISPR/Cas9 from the *Streptococcus* pyogenes is the most commonly used method in the genome editing process. CRISPR class II has a type V effector named Cpf1, which can be designed with highly specific CRISPR RNA to cleave corresponding DNA sequences [108,109]. Cpf1 was recently developed as a substitute to Cas9, because of its unique ability to target T-rich motives through staggered DSBs without the need to trans-activate crRNA. Cpf1 can also process RNA and the DNA nuclease operation. Studies have been conducted to examine the Cpf1 mechanism, aiming to achieve more precise DNA editing and to address it the crystal structure of Cas12b homologous [110]. Another study reported that small molecular compounds can enhance Cpf1 efficiency as they are directly involved in activating or suppressing signaling pathways for cellular repair. Thus, small-molecule–mediated DNA repair aids in useful CRISPR mediated knock-outs [111].

Unfortunately, the development of new crop varieties by genome editing has been delayed in many countries by strict GMO regulations across the globe. This is particularly true for areas obeying a process rather than a regulatory framework based on the product, like in the EU, where authorizations for new varieties developed by genome editing techniques are subject to time- and cost-intensive verification procedures [112]. A recent decision by the European Court of Justice announced the enforcement of strict GMO legislation on target genome editing tools, even if the product is entirely free of transgenes [113]. Process-based regulations were also introduced in at least 15 countries, such as Brazil, India, China, and Australia, while 14 countries, including Canada, Argentina, and the Philippines adopted a product-based regulation. Several countries still have no specific regulatory system in operation, including Paraguay, Myanmar, Chile, and Vietnam. One of the most interesting aspects of regulation is Argentina’s adoption, which is more versatile as it allows recent developments in genome editing to be taken into account. In the EU, genome-edited plants are typically listed as GMOs in compliance with the current legislation [114]. The control of genetically modified (GM) crops in the United States is authorized on a case-by-case basis, as set out in the structured framework for the control of biotechnology [115].

## 5. Speed Breeding (Time-Saving Tools) for Accelerating Plant Breeding

Most plant species create a bottleneck in their applied research and breeding programs, generating the need for technologies to accelerate up plant growth and generation turnover. In the early 1980s, NASA’s work was an inspiration for all plant scientists. In 2003, researchers at the University of Queensland coined the term “speed breeding” as a combination of methods developed to accelerate the speed of wheat breeding. Speed breeding protocols are currently being developed for several crops [16,30]. Speed breeding is suitable for diverse germplasm and does not require specific equipment for in vitro culturing, unlike doubled haploid (DH) technology, in which haploid embryos are produced to yield completely homozygous lines [116]. The principle behind speed breeding is to use optimum light intensity, temperature, and daytime length control (22 h light, 22 °C day/17 °C night, and high light intensity) to increase the rate of photosynthesis, which directly stimulates early flowering, coupled with annual seed harvesting to shorten the generation time [16,117]. Light intensity and wavelength plays a key in the regulation of flowering [118,119]. Croser et al. [120] developed early- and late-flowering genotypes for peas, chickpeas, faba beans, and lupins under controlled conditions using various parts of the light spectrum (blue and far red-improved LED lights and metal halide). These species showed a positive correlation to the diminishing red:far red-red proportion (R:FR). Accordingly, light with the most elevated power in the FR area is the most inductive [121,122]. In general, light with high R:FR (e.g., from fluorescent lamps) reduces stem enlargement and increases lateral branching, whereas light with a low R:FR (e.g., from incandescent lamps) strongly enhances stem elongation but inhibits lateral branching and flowering. This process is regulated by FR, while blue light mediates phytochrome FR (Pfr). Furthermore, the effect of R light on flowering repression is mediated by phytochrome R (Pr) [122,123].

Species-specific protocols to induce early flowering using certain environmental signals have been developed, such as short days or vernalization like RGA [48]. Greenhouse strategies under controlled conditions were compared with in vitro plus in vivo strategies and fast generation cycling by extended photoperiod [124,125,126]. The cost and space requirements associated with developing a large number of inbred lines can be reduced by implementing these practices in the breeding of small grain cereals grown at high densities (e.g., 1000 plants/m^2^) [127].

Until recently, speed breeding had been reported to shorten generation time by extending photoperiods (Figure 5), while certain crop species, such as radish (*Raphanus sativus*), pepper (*Capsicum annum*), and leafy vegetables such as Amaranth (*Amaranthus* spp.) and sunflower (*Helianthus annuus*) responded positively to increased day length [27,30,117,128]. Speed breeding of short-day crops has been limited because of their flowering requirements. Nevertheless, recently, Lee Hickey and his research team worked on developing protocols for short-day crop like sorghum, millet and pigeon pea with the International Crop Research Institute for the Semi-Arid Tropics (ICRISAT) as part of a project funded by the Bill and Melinda Gates Foundation (https://geneticliteracyproject.org/2020/03/02/how-speed-breeding-will-help-us-expand-crop-diversity-to-feed-10-billion-people/). Sorghum, millet, and pigeon peas are important plants for many smallholder farmers in Africa and Asia, refining protocols targeted for these types of users has significant implications for global subsistence agriculture. This goal involves improving the protocols and conditions required for the induction of early flowering and rapid crop development [117]. O’Connor et al. (2014) already reported successful results in the speed breeding of peanuts (*Arachis hypogaea*). Increased day length helped amaranth (*Amaranthus* spp.) to achieve more generations annually [129]. In staple food crops requiring shorter photoperiods to initiate the reproductive phases, such as rice (*Oryza sativa*) and maize (*Zea mays*), speed breeding can accelerate vegetative growth [130]. Using speed breeding, it is possible to develop successive generations of improved crops for field examination via SSD, which is cheaper compared to the production of DHs. Speed breeding is also favorable to gene insertion (common haplotypes) of distinct phenotypes followed by MAS of elite hybrid lines [31,131].

In conclusion, recent advances in plant breeding and genomics have contributed to the development of qualitatively and quantitatively improved cultivars. Innovative agronomic strategies, in addition to the usual practices, have led to remarkable agricultural outcomes. However, sustainable crop development to ensure global food security can only be achieved with the combined investments of private firms, extension workers, and the public sector.

## 6. Contribution of Plant Breeding to Crop Improvement

Molecular plant breeding was revolutionized in the 21^st^ century, leading to crop improvement based on genomics, molecular marker selection, and conventional plant breeding practices [10,39]. For instance, the average yield of wheat (*Triticum spp*.), maize (*Zea mays*), and soybean (*Glycine max*), all significant crops in the United States, showed a positive linear increase from 1930 to 2012 [132,133]. The introduction of recessive genes in off-season nurseries was commercialized by pioneering plant breeder Norman Borlaug (among others), which helped to reduce the time needed to develop new cultivars. For example, the time for developing a new wheat cultivar was reduced from 10–12 years to only 5–6 years [134].

In hybrid and pure line crop breeding, developing similar and homozygous lines is a time-consuming process. Cycle time has been reduced from five to two generations by producing homogeneous and homozygous lines using DHs in diverse crops [135,136]. The maize DH system is one of the most common, it uses the R1-NJ color marker. However, the DH system has various genotypic and biological limitations [136,137]. Different crop species show dependence on the genotype for haploid induction [138], adapting tissue culture (e.g., in case of anther culture), and chromosome doubling by colchicine [139]. Breeders using the DH system unintentionally practice many selections for loci, increasing the success rate of this approach [140], but this might limit genetic variation in the breeding populations in responsive genome regions. Another approach is the RNAi suppression of plant genes (for instance, the MutS HOMOLOG1 (*MSH1*) gene) in multiple plant species, which produces a variety of developmental modifications accompanied by adjustments in plant defense, phytohormones, and abiotic stress response pathways combined with methylome repatterning [141].

Although the evaluation and production of GM crops is an active area of research, this technology is currently restricted because of political and ethical concerns. Nevertheless, GMO technologies make use of the variations that are present in deliberately mutated or naturally occurring populations [39,142,143]. GMOs have a variety of practical applications, for instance, they can be used to produce plant proteins that are toxic to insect pests, create herbicide tolerance genes for weed control, and create “golden rice” biofortified with vitamin A [144]. The characterization and discovery of genes and promoters can offer precise and effective temporal and spatial control of the expression of different genes, which is crucial for the future use of GM crops [145].

The availability of published genome sequences for different crops is increasing every year, facilitated by the use of sequencing technologies that improve sequencing speed and cost [146,147]. Current sequencing technologies, such as the NGS technique, can sequence multiple cultivars with both small and large genomes at a reasonable cost [148]. Although various published genomes are considered to be incomplete, they remain a valuable tool to evaluate important crop traits such as grain traits, fruit ripening, and flowering time adaptation [83,149].

Modern plant breeding programs have engaged interdisciplinary teams with expertise in the fields of statistics, biochemistry, physiology, bioinformatics, molecular biology, agronomy, and economics [150]. Crop breeding has been revolutionized and research on the advancement of DNA sequencing technologies has started the “genomics era” of crop improvement [151]. The genomes of most of the essential crops have been sequenced, creating a much cheaper genotyping platform for DNA fingerprinting. SNPs are ubiquitous DNA markers in crop genomes, they are also cost-efficient and easy to handle. Therefore, in today’s crop improvement practices, genotyping large populations with a large number of markers is standard practice. Even whole-genome resequencing data are becoming easily available, giving unprecedented access to the structural diversity of crop genomes [65,83,152].

Currently, researchers are also using molecular genetic mapping of QTL of many complex traits vital in plant breeding. The detection and molecular cloning of genes underlying QTL enable the investigation of naturally occurring allelic variations for specific complex traits [85,153]. Plant productivity can be improved by identifying novel alleles through functional genomics or haplotype analysis. Advances in cereal genomics research in recent years have enabled scientists to improve the prediction of phenotypes from genotypes in cereal breeding [11,46].

Recently developed DNA-free CRISPR/Cas9 system delivery methods, different Cas9 variants, and RNA-guided nucleases offer new possibilities for crop genomic engineering [154]. The need to increase food security makes boosting crop production the primary objective of gene editing (Table 2). Crop yield is a complex trait that depends on several factors. The required phenotypes were found in plants with the loss of function mutations in yield related genes, highlighting the usefulness of CRISPR/Cas9 in improving yield-related traits by knocking out negative regulators affecting yield-determination factors, such as OsGS3 for grain size, OsGn1a for grain number; OsGW5, TaGW2, TaGASR7, and OsGLW2 for grain weight; TaDEP1 and OsDEP1 for panicle size, and OsAAP3 for tiller numbers. [155,156]. Similarly, three rice weight-related genes (GW5, GW2, and TGW6) were knocked out, causing pyramiding and increased weight [157]. The knockout of the *Waxy* gene using CRISPR/Cas9 resulted in the development of rice cultivars with higher nutritional quality [158]. DuPont Pioneer introduced a CRISPR/Cas9 knockout waxy corn line with high yields, ideal for commercial use [155]. A knockout of the *MLO* gene in tomato using CRISPR/Cas9 resulted in resistance to powdery mildew [159]. CRISPR has also been used to mutate the *OsERF922* transcription factor, resulting in resistance to rice blast, a destructive fungal disease [160].

By adopting a 22 h photoperiod and a temperature-controlled regime, generation times were considerably reduced in durum wheat (*T. durum*), spring bread wheat (*T. aestivum*), chickpea (*Cicer arietinum*), pea (*Pisum sativum*), barley (*Hordeum vulgare*), stiff brome (*Brachypodium distachyon*), canola (*Brassica napus*), and barrel clover (*Medicago truncatula*), compared with plants grown in a greenhouse with no supplementary light or those grown in the field. Under rapid growth conditions, plant development was normal, plants (such as wheat and barley) could be crossed easily, and seed germination rates were high [31,161,162,163].

## 7. Future Outlook

Although modern plant breeding relies on traditional techniques, the emergence of new approaches will undoubtedly increase its efficiency and effectiveness. In the future, we can expect a wide range of techniques to be developed using interdisciplinary principles to increase their benefits. Strategies for crop production, breeding methods, approaches to field testing, genotyping technologies, even equipment and facilities need to be implemented across crop species to keep our food, fiber and biobased economy diverse. The discovery of CRISPR/Cas9, CRISPR/Cpf1, base-editing, and RGA has revolutionized molecular biology and its innovative applications in agriculture, setting a turning point in plant breeding and cultivation. GMOs can positively disseminate a selectable gene across wild populations in a gene drive process.

Altogether, CRISPR-based gene drive systems will prove—in time—to be beneficial for mankind. They will, for instance, prevent epidemics, improve agricultural practices, and control the spread of invasive species as plant cultivars resistant to insect pests and pathogens and tolerant to herbicides are developed. Existing genome editing techniques can be improved with the help of speed breeding (e.g., genes responsible for late flowering could be knocked out using CRISPR/Cas9). After the successful transfer of Cas9 into the plant, the transgenic plant can then be grown under speed breeding conditions rather than the usual glasshouse conditions to obtain transgenic seeds as early as possible. Using this method, it is possible to obtain stable homozygous phenotypes in less than a year. Furthermore, this method also decreases generation time, as it normally takes several years to develop a GMO crop. However, more efficient breeding strategies combining these technologies could lead to a step-change in the rate of genetic gain. Therefore, CRISPR/Cas9, primarily based on genome editing and speed breeding, will likely gain in popularity. It will be a crucial technique to obtain plants with specific desirable traits and contribute to reaching our objectives for zero-hunger globally. The development of innovations is often applied to a few important economic crops, which require specific adaptation to the reproduction and propagation method and the "process" of the new line development for the various crops of interest. Likewise, the transition of technology usually originates in developed countries, mostly in the private sector. It should be transferred to the public sector and into the developing world, given the significant financial investment required for groundbreaking work.

## 8. Conclusions

The primary methods for crop improvement in modern agriculture are cross breeding, mutation breeding, and transgenic breeding. Such time-consuming, laborious, and untargeted breeding programs cannot satisfy the increasing global food demand. To deal with this challenge and to enhance crop selection efficiency, marker-assisted breeding, and transgenic approaches have been adopted, generating desired traits via exogenous transformation into elite varieties. These genome editing systems are excellent tools that provide rapid, targeted mutagenesis and can identify the specific plant molecular mechanisms for crop improvement. Crop breeding was revolutionized by the development of next-generation breeding techniques. Genome editing technologies have many advantages over traditional agricultural methods, given their simplicity, efficiency, high specificity, and amenability to multiplexing. We conclude that speed breeding, combined with genetic tools and resources, enable plant biologists to scale up their research in the field of crop improvement.

## Figures and Tables

**Figure 1 ijms-21-02590-f001:**
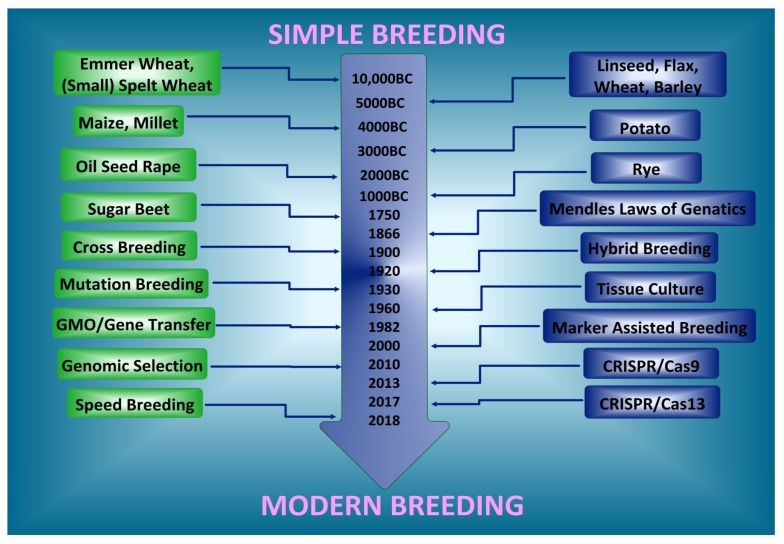
Historical milestones in plant breeding. For 10,000 years, farmers and breeders have been developing and improving crops. Presently, farmers feed 10 times more people using the same amount of land as 100 years ago.

**Figure 2 ijms-21-02590-f002:**
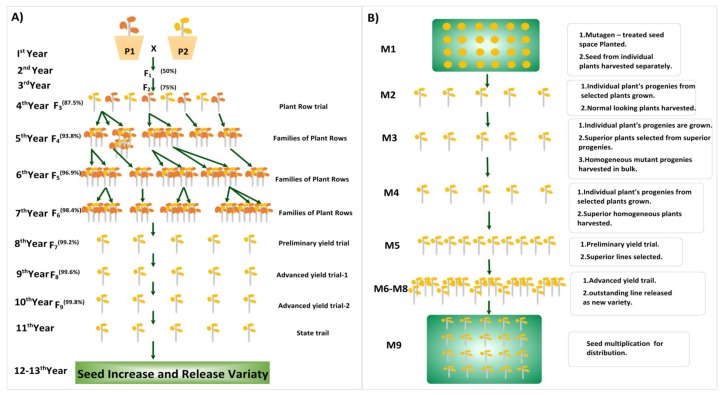
Improvement of agronomic traits using traditional breeding and chemical or physical mutagenic approaches. (**A**) Improving a trait (e.g., disease resistance) by the traditional breeding and for the introduction of the desired donor trait into the ‘chosen’ recipient line by selecting the progeny with the desired traits from the recipient line and crossing it with the donor line. (**B**) This process uses chemical or physical mutagens to generate mutants via random mutagenesis.

**Figure 3 ijms-21-02590-f003:**
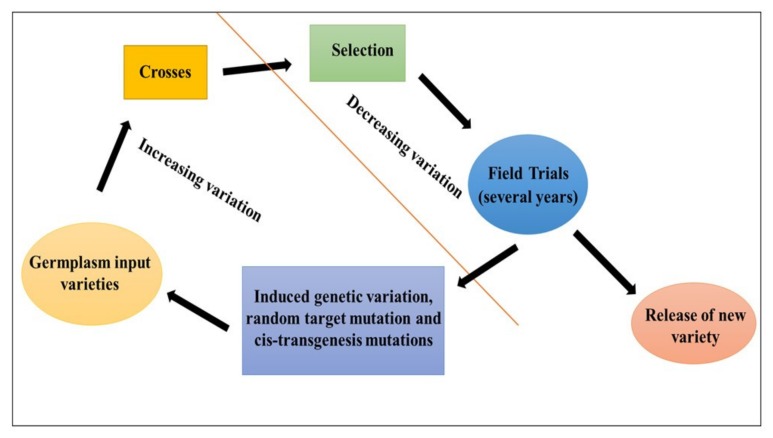
The plant breeding innovation cycle.

**Figure 4 ijms-21-02590-f004:**
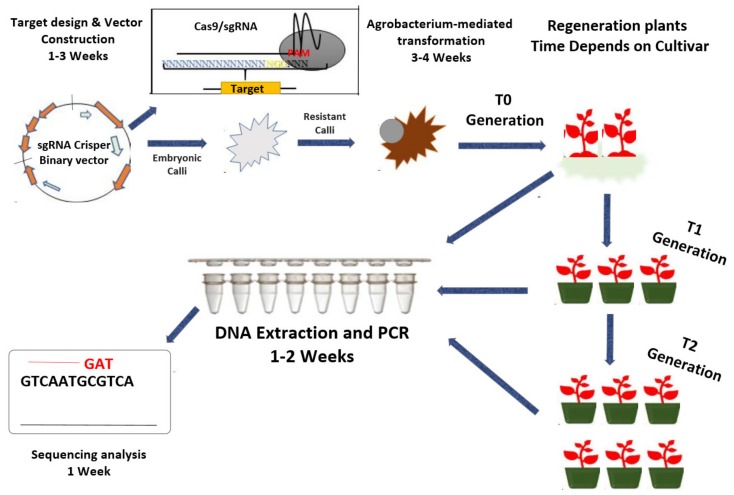
CRISPR–Cas9-based genome editing. CRISPR/Cas9 system uses Cas9 and sgRNA to cleave foreign DNA. It works in three steps: (1) the expression of the nuclear-localized Cas9 protein, (2) the generation of gRNA containing first 20-nt complementary to the target gene, and (3) the NGG PAM site recognition located nearly at the 3’ end of the target site. This process is followed by three additional steps: (1) design target and construction of a gene-specific sgRNA (vector), (2) CRISPR–Cas9 sgRNA can be transfected into the plant protoplast through *Agrobacterium*-mediated transformation, and (3) regenerated plants are screened for mutation via PCR-assay and sequencing. The estimated time needed is indicated for most steps.

**Figure 5 ijms-21-02590-f005:**
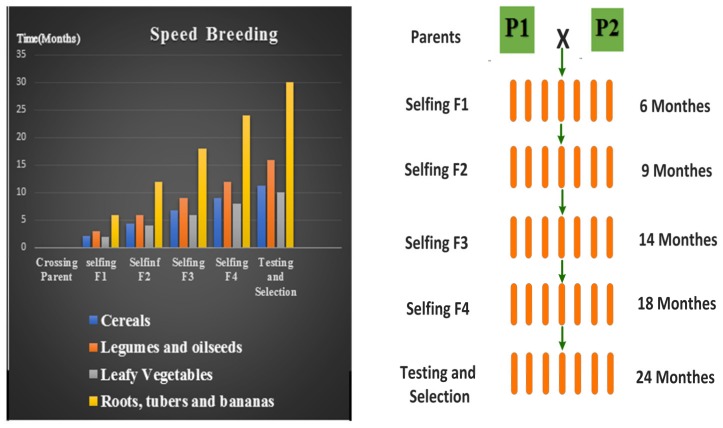
Graphical presentation of the elite line development procedure. Comparison of time (in months) required to develop elite lines from selected parents of some crops. Extended photoperiods induced earlier flowering and created 4 generations annually. The optimal temperature regime (maximum and minimum temperatures) should be applied for each crop. A higher temperature should be maintained during the photoperiod, whereas a fall in temperature during the dark period can aid in stress recovery. At the University of Queensland; (UQ), a 12-h 22 °C/17 °C temperature cycling regime with 2 h of darkness occurring within 12 h of 17 °C has proven successful. The figure is briefly modified from Watson et al. (2018).

**Table 1 ijms-21-02590-t001:** Examples of commonly used physical and chemical mutagens, their characteristics, and hazard impacts.

Types	Mutagens	Characteristics (Sources and Description)	Hazards	References
	X-rays	Electromagnetic radiation; penetrates tissues from just a few millimeters to many centimeters.	Dangerous, penetrating	[59]
	Gamma rays	60Co (Cobalt-60) and 137Cs (Caesium-137); electric magnet radiation generated with radiation isotope and nuclear reactors.	Dangerous, penetrating	[60,59]
**Physical Mutagens**	Neutron	235U; there are fast, slow, thermal types; formed in nuclear reactors; unloaded particles; penetrate tissues up to large numbers centimeter;	Very dangerous	[59,60]
	Beta particles	32P and 14C; reduced particle accelerators or radioisotopes; electrons; ionizing and penetrating tissues shallowly	Maybe dangerous	[60]
	Alpha particles	Sources originating from radiological isotopes; helium nucleus able to penetrate tissues heavily	Very dangerous	[59]
	Proton	Present in nuclear reactors and accelerators; derived from the nucleus of hydrogen; penetrate tissues up to several inches.	Very dangerous	[59,60]
	Ion beam	Positively charged ions are accelerated at a high speed and used to irradiate living materials, including plant seeds and tissue culture.	Dangerous	[60]
	Alkylating agents	The alkylated base can then degrade with bases to create a primary site which is mutagenic or recombinogenic or mispairs in DNA replication mutations, depending on the atom concerned.	Dangerous	[59]
	Azide	Just like alkylating agents.	Dangerous	[59]
	Hydroxylamine	Just like alkylating agents.	Dangerous	[59,56]
**Chemical Mutagens**	Nitrous acid	Acts through deamination, replacing cytosine with uracil, which can pair with adenine and thus result in transitions via subsequent replication cycles.	Very Hazard	[56]
	Acridines	Interspersing between the DNA bases, thus distorting the DNA double helix and the DNA polymerase, recognizes the new basis for this expanded (intercalated) molecule and inserts a frameshift in front of it.	Dangerous	[56]
	Base analog	Comprises the transformations (purine to purine and pyrimidine to pyrimidine) into DNA in place of the regular bases during DNA replication and tautomerizing (existent in two forms, which interconvert into one another such that guanine may be present in keto and enol forms).	Some may be dangerous	[56]

**Table 2 ijms-21-02590-t002:** Application of breeding techniques toward crop improvement.

Sr.no.	Species	Method	Traits	References
1	Rice	Cross Breeding	Increased spikelet number per panicle	[164]
2	Rice	Cross Breeding	Yield Increases	[165]
3	Wheat	Cross Breeding	Increase Grain Yield	[166]
4	Tomato	Mutation Breeding	Resistance to bacterial wilt *(Ralstonia solanacearum*)	[167]
5	Rapeseed	Mutation Breeding	Resistance to stem rot (*Sclerotinia sclerotiorum*)	[168]
6	Cotton	Mutation Breeding	Resistance to bacterial blight, cotton leaf curl virus	[169]
7	Barley	Mutation Breeding	Salinity tolerance	[170]
8	Sunflower	Mutation Breeding	Semi-dwarf cultivar/dwarf	
9	Cassava	Mutation Breeding	High-amylose content preferred by diabetes patients because it lowers the insulin level, which prevents quick spikes in glucose contents.	[171]
10	Groundnut	Mutation Breeding	Dark green, obovate leaf pod; increased seed size, higher yield, moderately resistant to diseases, increased oil and protein content	[172]
11	Maize	Transgenic Breeding	increased vitamin content (vitamins C, E, or provitamin A)	[173]
12	Tomato	Transgenic Breeding	Dry Matter Increases	[174]
13	Soybean	Transgenic Breeding	Altered carbohydrates metabolism	[174]
14	Barley	Molecular Marker	Adult resistance to stripe rust	[175]
15	Maize	Molecular Marker	Development of quality protein maize	[22]
16	Watermelon	Marker-Assisted Selection	Early Flowering	[176]
17	Canola	QTL	Dynamic growth QTL	[153]
18	Alfalfa	Intragenesis	Lignin content	[129]
19	Apple	Cisgenesis, Intragenesis	Scab resistance	[177,178]
20	Barley	Cisgenesis	Grain phytase activity	[179]
21	Durum wheat	Cisgenesis	Baking quality	[180]
22	Perennial ryegrass	Intragenesis	Drought tolerance	[181]
23	Poplar	Cisgenesis	Plant growth and stature, wood properties	[181]
24	Potato	Cisgenesis	Late blight resistance	[182]
25	Strawberry	Intragenesis	Gray mold resistance	[183]
26	Tomato	Gene editing/ZFN	Reduction of cholesterol and steroidal glycoalkaloids, such as toxic α-solanine and α- chaconine	[184]
27	Wheat	Gene editing/TALEN	Heritable Modification	[185]
28	Rice	Gene knockout/ CRISPR/Cas9	Fragrance	[186]
29	Bread Wheat and Maize	Gene knockout/ CRISPR/Cas9	Leaf development; Male fertility, Herbicide resistance	[187]
30	Poplar	Gene knockout/ CRISPR/Cas9	Lignin content; Condensed tannin content	[188]
31	Tomato	Gene editing/ CRISPR/Cas9	Leaf development	[189]
32	Soybean	Gene replacement/ CRISPR/Cas9	Herbicide resistance	[190]
33	Maize	Gene replacement/ CRISPR/Cas9	Herbicide resistance	[187]
34	Cotton	Genome Editing/ CRISPR/Cas9	Produce transgenic seeds without regeneration	[191]
35	Soybean	Genome Editing/ CRISPR/Cas9	Early Flowering	[192]
36	Rice	Genome Editing/ CRISPR/Cas9	Increased grain weight	[157]
37	Tomato	Genome Editing/ CRISPR/Cas9	Resistance to powdery mildew	[159]
38	Wheat	Gene knockout/ CRISPR/Cas9	low-gluten foodstuff	[193]
39	Rice	Gene knockout/ CRISPR/Cas9	Generate mutant plants which is sensitive to salt stress	[194]
40	Rapeseed	Gene knockout/ CRISPR/Cas9	Controlling pod shattering resistance in oilseed rape	[195]
41	Tomato, Potato	CRISPR/Cas9 Cytidine Base Editor	Transgene-free plants in the first generation in tomato and potato	[196]
42	Tobacco	Genome Editing /CRISPR/Cpf1	Plants harboring	[197]
43	Rice	Genome Editing /CRISPR/Cpf1	Regulate the stomatal density in leaf	[198]
44	Rice	Genome Editing /CRISPR/Cpf1	Stable mRNA equal	[100,199]
45	Maize	Genome Editing /CRISPR/Cpf1	Mutation frequencies doubled	[199]
46	Chickpea	Rapid generation advance (RGA)	Seven generations per year and enable speed breeding	[48]
47	Pea	Greenhouse strategy	6 Generation/year	[124]
48	Chickpea	Speed Breeding	4-6 Generation/year	[200]
49	Barley	Speed Breeding	Resistance to Leaf Rust	[16]
50	Spring wheat	Speed Breeding	Resistance to Stem Rust	[201]
51	Spring wheat	Speed Breeding	4-6 Generation/year	[16]
52	Barley	Speed Breeding	4-6 Generation/year	[16]
53	Peanut	Speed Breeding	2-3 Generation/year	[200]
54	Canola	Speed Breeding	4-6 Generation/year	[16]
55	Wheat	High-throughput phenotyping (HTP)	Development of improved, high-yielding crop varieties	[202]
56	Tomato	High-throughput phenotyping (HTP)	Using biostimulants to increase the plant capacity of using water	[203]

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
