# Peer review of "Conventional and Molecular Techniques from Simple Breeding to Speed Breeding in Crop Plants: Recent Advances and Future Outlook"

_ijms, 2020, doi:10.3390/ijms21072590_

Round 1

Reviewer 1 Report

Plant breeding, the application of genetics for the improvement of crops, is a very important and old activity of men to ensure and enlarge the supply of food for the world population. In this review, the authors summarize the main aspects and methods of plant breeding, including traditional breeding, artificial mutagenesis, molecular means and speed breeding. As a strong focus, the CRISPR/Cas9 technology is comprehensively presented along with its application on different plant species. A large table shows the crops and the methods used to improve specific traits.

The topic is relevant and of high importance. Many reviews were written about plant breeding in the past, but an overview about CRISPR and its impact in this applied research field is rare or maybe even missing. Of course it is ambitious to write a new review about plant breeding, and in fact this manuscript needs to be improved, since it is at various sites difficult to read and understand.

1-Structure
The introduction seems to be quite long, especially regarding the issues speed breeding and CRISPR. Here some aspect seems to be redundant, appearing here and again later in the other sections.
Chapter 2 (traditional/conventional breeding) includes High throughput phenotyping (HTP). Unfortunately, it is just mentioned that this is necessary and other references were given. However, most people in plant breeding meanwhile agree that the success will rely more on phenotyping methods than on genotyping methods in near future. This field of non-destructive phenotyping techniques with spectral sensors is dramatically increasing and should not be skipped.
The title of chapter 3 ‘Mutation via molecular tools’ do not fit to the following content. There are diagnostic molecular tools needed for the selection step in breeding, e.g. molecular markers for MAS/MAB, PCR for TILLING, NGS for e.g. marker development. These tools do not generate mutations but detect them. In contrast, the transformation (trans-/cisgenesis, ZFN, TALEN, CRISPR) in general creates mutations.
Chapter 4: In line 290 and line 296: This is a repetition, two times: transformation of explants via Agrobacterium
In line 319 the author mention that there are two classes of genome editing systems. Cpf1 belongs to class II, but what is class I and to which class belongs Cas9? If this information is relevant, start the chapter with this sentence to achieve a better structure.
Chapter 5: The term ‘plant breeding’ in the title is too general. I guess it is meant that the tools accelerate the time until flowering, i.e. the generation cycle. Since this chapter describes speed breeding, this term can also used in the title making it more precise.
The first sentence (line 330) is very general and fits to all chapters. Sentence 2 (line 331): What was NASA’s work? Is it that mentioned in line 90?
Chapter 7: The message of this section is very general. Are there any new technologies at the horizon? Are there limits in speed breeding or genome editing? Where is the next bottleneck?

2-I missed some general explanations e.g. that there are annual and perennial crops as well as selfing, outcrossing and asexual species. Therefore, the discussed methods are not always applicable to all crops. This can be mentioned in the introduction – with examples -, or later when the specific methods were described. For instance, long photoperiodic conditions in speed breeding are probably not applicable for crops that need short days; fruit crops with low numbers of seeds are less suitable for a mutagenesis approach like outlined in Fig. 2.

3-This review describes the achievements of genome editing in plant breeding and further chances for the future. However, this technology is differently regulated in the world, depending on public considerations and societal acceptance in the various countries. I missed a short paragraph about the regulation of CRISPR in some major countries (US, China, Europe, etc.). In addition, a thoughtful statement would be great whether ethical concerns may prevent or delay the application of genome editing and finally the development of plant breeding.

4-Language and writing style
The manuscript has to be checked and revised from an English native speaker. There are many words used in an inappropriate manner. For some sentences the meaning is not clear at all. Sometimes sentences are concatenated without any relation.
Line 20: maturation ???

Line 140: two times crossing/cross in this sentence: … can be improved by crossing …using cross-breeding ?!

Line 171: The sentence starts with the word ‘This’. To what object does it refers? To the preceding sentence?

Line 173: He examined these variations…: What is meant with these variations? No variations were mentioned in the sentences before.

Line 188: These mutations … What mutations? There are no mutations mentioned before

Line 248: there is no relation between these two sentences. Make new paragraf

Line 292: The mature plant is grown from … This makes no sense

Line 324/325: rephrase this sentence

Line 391: foundation ???

Line 396: rephrase – subordinate clause ????

Line 501: foundation ??

All figure legends and table captions have to be rephrased, since it is not English!!

Minor issues:

Line 80: milestones instead of millstones?

Line 160: Who is Prof. Detlef Weigel? Please add the place like : university xyz)

Line 227: … improved or new traits, …

Line 231: genotype instead of genotyping

Line 233: deciphering instead of sequencing

Line 267: check whether reference [14] is appropriate here, since this article is published in 2008, probably before the discovery of CRISPR

Line 269: please write Cas9 consistently either as Cas9 or Cas-9 (entire text)

Line 329: tools instead of tool

Figure 1: Please check the year for the first GM plant. I think it was 1983 and in 1996 the first GMO varieties were planted in the open field.

Figure 2: errors in right hand panel: plant’s progenies; delete dot after Superior; Advanced instead of Adsvanced

Table 1: Caption and throughout the headers and figures: check where capital or lower letters have to be used
Upper part, column Characteristics: check completely for language: very penetrating tissues ?; reduced in particle accelerators ?; 20%80% ?

Figure 4: is there one blue arrow missing between T0 and T1 generation?
Is the content of the box ‘Sequence 1 week’ correct?

Figure 5: what are leafly vegetables?

HTTP in the text, HTP in table 2: high-throughput trait phenotyping or high-throughput phenotyping?

GMO /GM: full description of abbreviation is not given

SB = ?

SVL = ?

UQ =?

Is reference [50] cited in the text?

Reviewer 2 Report

Dear Editor, I reviewed the manuscript ijms-728167 based on clarity of presentation and potential significance. I believe that this is well-written review where both the conventional and non-conventional plant breeding techniques are discussed. This paper is well designed and contains valuable information that must be of interest for readers of genetics and molecular plant breeding. The title properly reflects the subject of the paper. Figures, tables and manuscript structure are clear. The only problem I can found with this review are the abbrevations. For some well-known ones the full nemes are missing (e.g. NASA, EU, a non-GM path). The review is suitable for publication in IJMS.

Specific comments follow:

  1. Full name for CRISPR were used in Abstract Line 29, the full name for this abbreviation should be added once again in Introduction part.
  2. HTTP abbreviation is used in the text for high-throughput phenotyping. However in Table 2 Lines 56 and56 you used HTP.

Round 2

Reviewer 1 Report

The manuscript was revised very quickly. Some issues were met and improved the value of this review. E. g. the additional paragraph to the regulation of the CRISPR technology (remark 8).

However, I am afraid to say that I have still severe concerns about the weak English and that I was indeed somehow disappointed. I guess the point is that the authors are not so well in writing in English including the use of the scientific terminology. On the other hand, the person, who probably was not familiar with plant science, did the professional English editing. Anyway, the revision was not carefully done, since there are still many sentences that lack sufficient language quality needed to understand the paper (very long sentences, incomplete sentences, non-existing English words, mode of expression, etc.). I have especially difficulties with the revisions almost all done in blue color. Maybe this was done by a different person than that what is written in red color.

It would be pity if the text could not be improved significantly. This is a condition to be published in an excellent journal like IJMS which has a high impact factor.

Next to the major issue of language here are some detailed comments:

Remark 2: Line 227-241: I can just guess what you want to express but this not English. Please take time for phrasing and make full sentences. Alternatively, delete this part.

Remark 10: Now the sentence is incomplete. The issue I wanted to draw attention to is that you twice mentioned cross breeding. First, you explained how a crossing works and then you added ‘using cross breeding’. You can make a period after ‘ … trait’ and delete the rest.

Remark 11: It is not better. Now you have two times the word ‘plant’ in the sentence in line 184-185. ‘Mutant crop plants’ should mean ‘mutated crop plants’? Please specify the crop and add the references for these reports.
Line 185-186: The word ‘this’ still implies that you mean a specific mutation. Which mutation was identified (color trait, height, etc.)? It sounds not reasonable that Hugo de Vries identified in the 19th the mutation of a crop plant in China of 300 BC. I still have no idea about the link between these two sentences.

Remark 13: Now the sentence makes no sense, since mutagens can not be induced. They are inducing means. I would suggest: ‘They can induce mutations in almost all . . . ‘

Remark 15: Line 322: Seedlings arise from seeds not from calli! And what means ‘mature seedling’? Do you mean ‘regenerated plantlets’?

Remark 17: What is meant with the term ‘organizations’ in this context? Could you give an example of an organization?

Remark 18: ‘was’ instead of ‘is’. Line 478: no full sentence

Remark 20: There is hardly an improvement of the English after the rephrasing.

Remark 30: ‘plants progenies’? Do you mean plant’s progenies? Please check

Remark 33: I checked the term ‘leafly’ on my own: it does not exist. Do you mean ‘leafy’?

Line 160: precision not Precision

Line 167: de novo in italic

Line 168: 7 x 10-9 was in the first version the correct type of writing

Line 359: Is it Streptococcus pyogene or Streptococcus pyogenes?

Line 373-378: Very long sentence. Split it. Areas means here countries?

Author Response

Dear Editor,

Thanks for forwarding the reviewer’ comments and giving us a chance to improve our manuscript. We have read the comments carefully and revised our manuscript accordingly. Responses to reviewers’ comments is given below.

Thank you for spending time on our manuscript, please feel free to contact us if any problem still exists.

Yours sincerely,

Authors: Sunny Ahmar, Rafaqat Ali Gill, Ki-Hong Jung, Aroosha Faheem, Muhammad Uzair Qasim, Mustansar Mubeen and Weijun Zhou.

Response to Reviewers’ Comments to Authors

Reviewer 1:

Reviewer’s comments

Remark 1: However, I am afraid to say that I have still severe concerns about the weak English and that I was indeed somehow disappointed. I guess the point is that the authors are not so well in writing in English including the use of the scientific terminology. On the other hand, the person, who probably was not familiar with plant science, did the professional English editing. Anyway, the revision was not carefully done, since there are still many sentences that lack sufficient language quality needed to understand the paper (very long sentences, incomplete sentences, non-existing English words, mode of expression, etc.). I have especially difficulties with the revisions almost all done in blue color. Maybe this was done by a different person than that what is written in red color.

It would be pity if the text could not be improved significantly. This is a condition to be published in an excellent journal like IJMS which has a high impact factor.

Response: Thank you for your valuable comments and sorry for inconvenience. We have revised the MS for grammatical errors as well as recheck according to the standard for IJMS. Furthermore, we have done English proofreading again by English editing services from native professional speaker with special subject in Plant Sciences.

Remark 2: Remark 2: Line 227-241: I can just guess what you want to express but this not English. Please take time for phrasing and make full sentences. Alternatively, delete this part.

Response:  Thanks for your kind comments and suggestions. We have carefully gone through and according to your suggestion we have deleted this part.

Remark 3: Remark 10: Now the sentence is incomplete. The issue I wanted to draw attention to is that you twice mentioned cross breeding. First, you explained how a crossing works and then you added ‘using cross breeding’. You can make a period after ‘ … trait’ and delete the rest.

Response: Thanks for your kind reminder again. We have rephased the sentence in lines 162-163.

Remark 4: Remark 11: It is not better. Now you have two times the word ‘plant’ in the sentence in line 184-185. ‘Mutant crop plants should mean ‘mutated crop plants’? Please specify the crop and add the references for these reports.

Response: Thanks for your kind notice. We have revised these lines and rephased the sentence in lines 197-202 for better understanding and also cite the reference for detailed review. Actually, we are discussing about history of plant mutation not a specific plant, we have revised this paragraph.

Remark 5: Line 185-186: The word ‘this’ still implies that you mean a specific mutation. Which mutation was identified (color trait, height, etc.)? It sounds not reasonable that Hugo de Vries identified in the 19th the mutation of a crop plant in China of 300 BC. I still have no idea about the link between these two sentences.

Response: We have revised these lines and rephased the sentence in lines 201-204 for better understanding and also cite the reference for detailed review.

Remark 6: Remark 13: Now the sentence makes no sense, since mutagens cannot be induced. They are inducing means. I would suggest: ‘They can induce mutations in almost all . . . ‘

Response: Thank you for your valuable comments. We have revised these lines 219-220. Please see the revised version.

Remark 7: Remark 15: Line 322: Seedlings arise from seeds not from calli! And what means ‘mature seedling’? Do you mean ‘regenerated plantlets’?

Response: Thanks for your kind notice. Yes, that means ‘regenerated plantlets’ and we have replaced this word with mature seedling in line 331.

Remark 8: Remark 18: ‘was’ instead of ‘is’. Line 478: no full sentence

Response: Thanks for your kind notice. We have improved the stated problem line 463.

Remark 9: Remark 20: There is hardly an improvement of the English after the rephrasing.

The manuscript has to be checked and revised from an English native speaker. There are many words used in an inappropriate manner. For some sentences the meaning is not clear at all. Sometimes sentences are concatenated without any relation.

Response: Thank you for your kind suggestions and sorry for inconvenience. We have revised the MS for grammatical errors as well as recheck according to the standard for IJMS. Furthermore, we have done English proofreading again by English editing services from native professional speaker with special subject in Plant Sciences as well as we have made the correction and rephased figures legends and tables and make it easy to understand for readers.

Remark 9: Remark 30: ‘plants progenies’? Do you mean plant’s progenies? Please check

Response: Thanks for your comments. We have made correction in figure 2 as plant’s progenies.

Remark 10: Remark 33: I checked the term ‘leafly’ on my own: it does not exist. Do you mean ‘leafy’?

Response: Thanks for your kind notice. Yes, it means leafy, we have made the correction according to your suggestion in revised version in text as well as in Figure 5, line 424.

Remark 11: Line 160: precision not

Response: Thanks for your valuable comments. We have made the correction in line 171.

Remark 12: Line 167: de novo in italic

Response: Thanks for your valuable comments. We have made the correction in line 181.

Remark 13: Line 168: 7 x 10-9 was in the first version the correct type of writing

Response: Thanks for your kind suggestion. We have made the correction in line 181.

Remark 14: Line 359: Is it Streptococcus pyogene or Streptococcus pyogenes?

Response: Thanks for your kind notice. We have made the corrections in line 353. Please see revised MS.

Remark 15: Line 373-378: Very long sentence. Split it. Areas means here countries?

Response: Thanks for your kind notice. We have rephased the sentence in lines 370-375 and make it easy to understand. Yes, areas mean countries we have corrected.

Remark 16: Remark 17: What is meant with the term ‘organizations’ in this context? Could you give an example of an organization?

Response: Thanks for your kind notice. We have improved our sentences in line 463.

Round 3

Reviewer 1 Report

Due to the substantial revision of the manuscript it is now much better and more fluent to read. Good work!

I would now recommend this manuscript for publication without any further review. There are only some very minor issues I noticed while reading the text that can be easily checked before generating the proofs:

L 172: unintended line break ?

L 173: applicable – line break within the word ?

L 182: I would say it should be ‘genomes’ instead of ‘genome’ (like it was in the first version). Please check

L 183: My suggestion for this sentence: ‘This is expected to be true for the genomes of most other plant species.’

L 185 + L 198: Professor / professor; please write uniformly

L 214: this is difficult to understand. My suggestions: ‘ . . .  ‘chosen’ recipient line by selecting the progeny . . . ‘ or ‘ . . .  ‘chosen’ recipient line, which requires the selection of the progeny . .  . ‘

L 220: planting

L 225: This is a long sentence with 8 commas! I suggest to make a period after ‘. . . single cells.’ and delete the rest of the sentence. Alternatively, I would insert the part ‘and other forms of in vitro cultured plant tissues’ after ‘…vegetative propagated plants, …’

L 232: hazards impacts ? My suggestion: impacts of hazards or hazard impacts

L 245: programs instead of programmers

L 448: I am not sure whether this sentence is correct. Could you please check it again?
‘Data was taken from Watson et al. (2018) using speed breeding (20 months, 3-5 generations).’

L 626 and throughout the text: RDA = Rapid Generation Advance. Is this the correct abbreviation? In the reference [48] you cited, it is abbreviated with RGA

Author Response

Dear Editor,

Thanks for forwarding the reviewer’ comments and giving us a chance to improve our manuscript. We have read the comments carefully and revised our manuscript accordingly. Responses to reviewers’ comments is given below.

Thank you for spending time on our manuscript, please feel free to contact us if any problem still exists.

Yours sincerely,

Authors: Sunny Ahmar, Rafaqat Ali Gill, Ki-Hong Jung, Aroosha Faheem, Muhammad Uzair Qasim, Mustansar Mubeen and Weijun Zhou.

Response to Reviewers’ Comments to Authors

Reviewer 1:

Remark 1: L 172: unintended line break?

Response: Thank you for your kind notice. We have checked the line break and corrected it (line 147-148).

Remark 2: L 173: applicable – line break within the word?

Response: Thank you for your kind notice. We have checked the word break and corrected it (line148).

Remark 3: L 182: I would say it should be ‘genomes’ instead of ‘genome’ (like it was in the first version). Please check

Response: Thank you for your suggestion. We have correct genome with genomes in line 154.

Remark 4: L 183: My suggestion for this sentence: ‘This is expected to be true for the genomes of most other plant species.’

Response: Thank you for your suggestion. We have replaced sentence according to your suggestion in line 155-156.

Remark 5: L 185 + L 198: Professor / professor; please write uniformly

Response: Thank you for kind notice. We have written uniformly in line 157 and 168 of revise MS.

Remark 6: L 214: this is difficult to understand. My suggestions: ‘ . . .  ‘chosen’ recipient line by selecting the progeny . . . ‘ or ‘ . . .  ‘chosen’ recipient line, which requires the selection of the progeny. .  . ‘

Response:  Thank you for your valuable suggestion. We have updated according to your suggestion in line 180.

Remark 7: L 220: planting

Response: Thank you for your kind notice. We have updated (line 186).

Remark 8: L 225: This is a long sentence with 8 commas! I suggest to make a period after ‘. . . single cells.’ and delete the rest of the sentence. Alternatively, I would insert the part ‘and other forms of in vitro cultured plant tissues’ after ‘…vegetative propagated plants, …’

Response:  Thank you for your valuable comment. We have corrected in line 189-190.

Remark 9: L 232: hazards impacts? My suggestion: impacts of hazards or hazard impacts

Response:  Thank you for your suggestion. We have corrected the sentence in line 196.

Remark 10: L 245: programs instead of programmers

Response: We have replaced programs with programmers(line 207).

Remark 11: L 448: I am not sure whether this sentence is correct. Could you please check it again?

‘Data was taken from Watson et al. (2018) using speed breeding (20 months, 3-5 generations).’

Response: Thank you for your response. We have checked again and corrected it (line 394-395).

Remark 12: L 626 and throughout the text: RDA = Rapid Generation Advance. Is this the correct abbreviation? In the reference [48] you cited, it is abbreviated with RGA

Response: Thank you for your kind notice. We have replaced RDA with RGA in whole MS.